# Computer-Aided Histopathological Characterisation of Endometriosis Lesions

**DOI:** 10.3390/jpm12091519

**Published:** 2022-09-16

**Authors:** Brett D. McKInnon, Konstantinos Nirgianakis, Lijuan Ma, Carlos Alvarez Wotzkow, Selina Steiner, Fabian Blank, Michael D. Mueller

**Affiliations:** 1Department of Gynecology and Gynecological Oncology, Inselspital, Bern University Hospital, University of Bern, Friedbuehlstrasse 19, 3010 Bern, Switzerland; 2Department of BioMedical Research, Live Cell Imaging, University of Bern, 3010 Bern, Switzerland

**Keywords:** endometriosis, pain, stromal, epithelial, CD10, cytokeratin, subtype, Qupath, histopathology

## Abstract

Endometriosis is a common gynaecological condition characterised by the growth of endometrial tissue outside the uterus and is associated with pain and infertility. Currently, the gold standard for endometriosis diagnosis is laparoscopic excision and histological identification of endometrial epithelial and stromal cells. There is, however, currently no known association between the histological appearance, size, morphology, or subtype of endometriosis and disease prognosis. In this study, we used histopathological software to identify and quantify the number of endometrial epithelial and stromal cells within excised endometriotic lesions and assess the relationship between the cell contents and lesion subtypes. Prior to surgery for suspected endometriosis, patients provided menstrual and abdominal pain and dyspareunia scores. Endometriotic lesions removed during laparoscopic surgery were collected and prepared for immunohistochemistry from 26 patients. Endometrial epithelial and stromal cells were identified with Cytokeratin and CD10 antibodies, respectively. Whole slide sections were digitised and the QuPath software was trained to automatically detect and count epithelial and stromal cells across the whole section. Using this classifier, we identified a significantly larger number of strongly labelled CD10 stromal cells (*p* = 0.0477) in deeply infiltrating lesions (99,970 ± 2962) compared to superficial lesions (2456 ± 859). We found the ratio of epithelial to stromal cells was inverted in deeply infiltrating endometriosis lesions compared to superficial peritoneal and endometrioma lesions and we subsequently identified a correlation between total endometrial cells and abdominal pain (*p* = 0.0005) when counted via the automated software. Incorporating histological software into current standard diagnostic pipelines may improve endometriosis diagnosis and provide prognostic information in regards to severity and symptoms and eventually provide the potential to personalise adjuvant treatment decisions.

## 1. Introduction

Endometriosis is a common benign gynaecological disease characterised by the growth of endometrial tissue outside the uterine cavity and has a diverse clinical presentation. It is associated with subfertility [1] and painful symptoms, including chronic pelvic/abdominal pain, menstrual pain, dyspareunia, dysuria, and dyschezia [2]. The most commonly accepted theory of endometriosis pathogenesis is that of retrograde menstruation in which viable endometrial cells are refluxed back into the peritoneal cavity during menstruation. These cells adhere to the underlying tissue, establish lesions, and proliferate in response to hormonal and inflammatory signals [3]. 

Treatment requires the surgical removal of lesions and adjuvant hormonal suppression to avoid disease and symptom recurrence [4]. Endometriotic lesions are heterogenous in appearance, varying in their size, shape, colour, and location [5], and are currently split into three major subtypes. These subtypes from increasing severity include superficial peritoneal lesions (SUP), endometrioma (OMA), and deeply infiltrating endometriosis (DIE), the most severe form characterised by infiltration of more than 5 mm into the underlying tissue [6]. Diagnosis is via visualisation during surgery with histopathological confirmation of the presence of endometrial cells in the excised tissue. Currently, the histopathological diagnostic paradigm provides no insight into disease characteristics or the need for appropriate adjuvant treatment. A more comprehensive histological characterisation may assist treatment decisions. 

To better relate the lesion to patient outcomes and refine diagnostic methods, an automated characterisation of the lesion could provide great benefit. Recent advances in digital pathology provide the opportunity to automatically detect, quantify, and characterise cells within lesion tissue to quickly and accurately assess endometriosis foci. Using specific cytokeratin and CD10 antibodies to identify endometrial epithelial and stromal cells, respectively, in suspected endometriosis tissue excised during surgery, we trained Qupath histopathology software to automatically detect and quantify endometrial epithelial and stromal cells in endometriotic lesions. Using this technology, we wished to determine whether automated histological software could provide additional data to aid the diagnosis and prognosis of endometriosis.

## 2. Material and Methods

### 2.1. Sample and Pain Collection

Prior to surgery informed consent was obtained from all patients and ethics approval granted by the local ethics committee. Endometriotic lesions were collected from women undergoing laparoscopic surgery for reasons of idiopathic infertility or chronic pelvic pain. This study was approved by the Bernese Cantonal Ethical Review Board (149-03) and informed consent collected from all patients. Prior to surgery, patients were asked to provide details on painful symptoms, including menstrual pain, abdominal pain, and dyspareunia, via a visual analogue scale from 0 to 10, with 0 being the lowest and 10 being the most severe pain.

During laparoscopic surgery, endometriotic lesions were removed and their location noted. Lesions were classed as either (i) SUP lesions, (ii) OMA, or (iii) DIE [6]. Removed tissue was sent for histological diagnosis by a trained pathologist and any remaining tissue fixed in formalin for subsequent immunohistochemistry and analysis via Qupath. Clinical characteristics, including age, body mass index (BMI), and hormonal treatment prior to surgery, were collected.

### 2.2. Immunodetection of Epithelial and Stromal Cells in Excised Lesion Tissue

Samples from ectopic lesions were formalin fixed for 4 h and embedded in paraffin. Samples were sectioned at 10 μm, mounted on glass slides, dewaxed in xylene, and rehydrated through a series of increasing ethanol concentrations. Non-specific binding was blocked by incubation in 3% bovine serum albumin (BSA) in tris-buffered saline (Tris 100 mM, NaCl 0.15 M; pH7.4) for 30 min. Sections were incubated with either mouse anti-cytokeratin antibody (Novus Biologicals, Littleton, CO, USA, 1:200 dilution) or a rabbit anti-CD10 antibody (Novus Biological, 1:100 dilution) diluted in tris buffer containing 3% BSA in a humidified chamber overnight. Slide were washed with tris-buffered saline containing 0.1% Tween 20 and incubated with either biotinylated goat anti-mouse antibody (Dako, Glostrup, Denmark, dilution 1:200) or swine anti-rabbit (Dako, dilution 1:200) for 90 min at room temperature. After washing, the sections were incubated with avidin-biotin-horseradish peroxidase complex (Vectastain ABC Kit, Vecter laboratories, Newark, NJ, USA) for 45 min, with the detection of bound antibody performed with diaminobenzidine substrate. Slides were counter stained with haematoxylin and mounted in Aquatex (Merk, Darmstadt, Germany).

### 2.3. Tissue Section Digitisation and Automated Cell Detection

Whole slide images were scanned and digitised on the Aperio Digital pathology slide scanner (Leica Biosystems, Nussloch, Germany) and images analysed with the digital pathology software QuPath [7]. To perform automated detection of epithelial and stromal cells, we first trained an object classifier: a function that allows automatic identification of specific objects or cell types within the image based on an initial user identification and classification. Training of the object classifier allowed the subsequent assessment of 43 predefined parameters (Appendix A). Using the object classifier provided the opportunity to apply these parameters, in addition to the antibody labelling, to all cells, thereby identifying any epithelial or stromal cells across the entire slide. This includes the possibility to include cells that may have been labelled negative for the marker but were sufficiently similar on the other 43 parameters.

To train the object classifier, we used 5 randomly selected slides that were incubated with either cytokeratin or CD10 antibodies. We manually identified epithelial and stromal cells, based on the antibody labelling in at least 5 regions in each slide. In both sets of slides, we also anointed non-lesion regions. The variability and accuracy of the object classifier was assessed after the use of 1 to 5 slides for training by comparing the number of cells assigned to each category and visual inspection of the cell detection consistency. This was performed independently for both the cytokeratin- and CD10-labelled slides to create independent object classifiers for both cell types.

Once trained, object classifiers were applied to each slide within the cohort. An initial visual inspection of each slide was performed to exclude regions due to technical issues, such as tissue folds or high non-specific background labelling. Cell counting was subsequently performed as an unsupervised, automated process with the relevant object classifier. Both epithelial and stromal cells were counted as were the number of cells that showed either negative, mild (1+), moderate (2+), or heavy (3+) immunoreactivity.

### 2.4. Statistical Analysis

To compare two quantitative variables, a parametric *t* test was used. If more than two quantitative variables were present, an ordinary one-way analysis of variance (ANOVA) was applied. For all tests, we performed the Pearson’s parametric correlation coefficient and a *p* value < 0.05 was considered significant. All statistical analysis was performed using Graph Pad Prism (Version 8.4.2) Dotmatics, Boston, USA. To determine the relative cell content, a stromal to epithelial cells ratio was determined. To determine age, BMI, and average pain data, if a patient had multiple lesions, this patient-level data was included only once. To split this data by lesion subtype, it was assigned to the most severe lesion, as described previously [6].

## 3. Results

### 3.1. Patient Data

We collected 31 endometriotic lesions from 26 individual patients. A single lesion was collected from 23 patients, 2 lesions were collected from one patient both of which were DIE, and 3 lesions were collected from an additional 2 patients. One of these patients had all SUP lesions collected, whereas the other patient had one OMA and two DIE lesions. In total, all lesions collected included 9 SUP, 10 OMA, and 11 DIE, with one lesion unable to be assigned to a location. The revised American Fertility Score (rAFS) of the 26 women included 11 with stage IV, 10 with stage III, 3 with stage II, and 2 women with stage I. In this cohort, 11 patients were receiving no hormonal treatment prior to surgery, 8 patients were receiving oral contraceptives, and 3 patients reported GnRHa usage. In the remaining four patients, we were unable to confirm their treatment history. To determine the menstrual stage at which lesions were removed, we relied on self-reported cycle day. Women taking hormonal treatments (*n* = 11) were considered amenorrhoeic and 5 women provided sufficient information to be considered post-luteal stage, with no information available for the remaining 10 women.

The average age for all patients was 32.26 ± 0.84 and the average BMI was 21.99 ± 0.59 (Table 1). Menstrual pain values were provided by 17 patients, with the remaining indicating they were either not experiencing pain, which was recorded as 0, or were not provided and were excluded from the analysis. The average value for menstrual pain was 6.16 ± 0.79. Abdominal pain scores were received from 22 patients (2.12 ± 0.46). Dyspareunia scores were also received from 22 patients (1.89 ± 0.57).

### 3.2. Automated Endometrial Stromal Cell Detection and Quantification in Excised Tissue

We applied CD10 labelling (Figure 1A) for stromal cell identification and trained an object classifier to perform an automated slide-wide stromal cell detection (Figure 1B). Using pixel smoothing to delineate regions of stromal cell positivity, we created predictive models to delineate endometriotic lesion borders and quantitate stromal cell content (Figure 1C). Automated cell counting identified 14,158 ± 3583 (mean ± standard error of the mean (SEM)) stromal cells per section. Most cells were considered to have heavy labelling ((3+) (5800 ± 1266)) and moderate labelling ((2+) 4983 ± 1548), with light labelling (1+) the least commonly observed (2971 ± 893). Interestingly, 404 ± 127 stromal cells per section that were negative for CD10 labelling were identified as stromal cells using the other parameters defined in the object classifier (Figure 1D).

Automated stomal cell detection and quantification were compared in lesions from different locations. SUP lesions (5979 ± 3036) (Figure 2A) contained the least amount of stromal cells followed by OMA (Figure 2B) (12,542 ± 4824), with DIE containing the most (23,296 ± 8485) (Figure 2C). This difference, however, did not reach significance (Figure 2D). A comparison of the heavily labelled CD10 cells (3+) revealed a significant difference (*p* < 0.05) between DIE lesions (99,970 ± 2962) and SUP lesions. (2456 ± 859) (Figure 2E).

### 3.3. Automated Epithelial Cell Detection and Quantification in Excised Tissue

Using the cytokeratin-positive cells to identify endometrial epithelial cells (Figure 3A), we identified epithelial cells in both characteristic glandular structures and without characteristic glandular structures. Using the object classifier, we performed automated cell quantification (Figure 3B). Pixel smoothing assisted in defining endometriotic foci boundaries and variation in the staining intensity within the boundaries (Figure 3C). The automated cell detection and quantification identified a mean and SEM value of 18,426 ± 3421 endometriotic epithelial cells per section (Figure 3D). This included 6169 ± 1786 epithelial cells with light labelling (1+), 5636 ± 1011 cells with moderate labelling (2+), and 6578 ± 922 cells with heavy labelling (3+). The object classifier identified 43 ± 23 cells that were negative for cytokeratin staining but identified as epithelial based on the other parameters. There was no variation in the number of light, moderate, or heavily labelled cells.

Comparison of the epithelial cells in different lesion subtypes indicated both SUP (12,598 ± 3570) (Figure 4A) and OMA (12,249 ± 3014) (Figure 4B) lesions had similar numbers of epithelial cells while DIE lesions had the highest mean number of cells (25,290 ± 7519) (Figure 4C), although this difference was not significant (Figure 4D). A comparison of cytokeratin-positive cells between lesion subtypes based on labelling intensity also found no difference between epithelial cell content in SUP, OMA, or DIE.

### 3.4. Comparison between the Epithelial and Stromal Contents in Lesions

Using the automated cell counts, we compared the ratio of epithelial and stromal cells in each of the endometriotic subtypes. There was a higher mean number of epithelial cells (18,426 ± 3421) compared to stromal cells (14,013 ± 3468) when assessing all lesions (Figure 5A), although these differences were not significant. We compared the ratio of stromal to epithelial cells between lesions of different subtypes and found this ratio was only above 1 in DIE lesions (1.225 + 0.4107), suggesting a large stromal to epithelial content in this lesion subtype. Both SUP (0.4278 ± 0.1147) and OMA (0.9327 ± 0.2829) ratios were less than 1, suggesting a greater epithelial content in these lesions. A comparison between these ratios did not reach significance (Figure 5B). 

### 3.5. Influence of Hormonal Treatment on Endometriotic Lesion Cells and Their Ratio

We wished to determine whether hormonal treatments have a direct effect on the cell content of endometriotic lesions. A comparison of epithelial (Figure 6A) and stromal cell (Figure 6B) counts between samples excised from women who received hormonal treatment compared to those who did not showed no significant difference in epithelial cells from tissue excised from women who had not received treatment (18,338 ± 5968) compared to those who had (14,267 ± 3430). Similarly, although a lower number of endometrial stromal cells were found in hormonal-treated samples (8859 ± 2615), compared to untreated samples (13,381 ± 4705), it was not significant. 

Stratifying the treatment condition by hormonal preparations, we split hormonal treatment into either the oral contraceptive pill (OCP) (*n* = 12) or gonadotropin-releasing hormone analogue (GnRHa) (*n* = 3). There was no significant difference in the pain reported by pateints based on hormonal preparation (Appendix A). Although the mean epithelial cell counts in the no treatment groups were higher (18,338 ± 5968) compared to both OCP (16,548 ± 4013) and GnRHa (5143 ± 2349), the difference was not significant (Figure 6C). More stromal cells were found in the no treatment group (13,381 ± 4705) compared to OCP (7097 ± 2627), although GnRHa showed the highest number (15,908 ± 7511); however, again, no significant difference was observed (Figure 6D). 

### 3.6. Association between Automated Cell Counts, Stage of Disease, and Patient Symptoms

Finally, we compared automated cell quantification values to patient symptoms recorded prior to surgery and endometriosis staging of patients recorded during surgery. We found no association between stage of disease, separated into either mild (rAFS stage I–II) or (severe III–IV) for either epithelial cells, stromal cells, or cells combined (Table 2). We did, however, identify a significant positive correlation (*p* < 0.05) between stromal cells and abdominal pain and epithelial cells and abdominal pain (*p* < 0.01). Similarly, for the total number of cells (stromal and epithelial), the correlation with abdominal pain was increased (*p* < 0.001) (Table 3). No significant correlation was observed with either menstrual pain or dyspareunia with either the stromal cells, epithelial cells, or total cells together (Table 3). 

## 4. Discussion

Endometriosis is an enigmatic and painful disease that affects over 10% of women of reproductive age [8]. Endometriotic lesions are extremely heterogenic and the relationship between lesion appearance and symptoms is not well characterised [9]. In this study, we applied automated histopathology software and immunohistochemistry to automatically quantify the endometrial epithelial and stromal cells in excised tissue from different endometriosis subtypes. We quantitated the endometrial cellular content of endometriosis foci, the relationship between epithelial and stromal cells in lesion subtypes, the influence of hormonal treatments, and the relationship with clinical symptoms. 

Laparoscopic excision and histopathological confirmation of endometrial tissue remains the gold standard for endometriosis diagnosis and treatment [4]. Subsequent studies using the excised tissue, aiming to identify a molecular relationship between clinical symptoms, are common but have shown little relationship between physical characteristics and the pain reported [10]. These studies are most likely hindered by the variable extent of endometriosis in the excised tissue. Lesion size, number of foci, and cell composition may vary significantly from one piece of tissue to the next and, hence, the size of the tissue excised may have little reflection on the extent of the actual endometriosis. Using cytokeratin and CD10 as markers of epithelial and stromal cells, respectively, and as the basis for training the QuPath software for automated identification of epithelial and stroma cells, we quantified these populations within each tissue section.

Our analysis showed a greater proportion of epithelial cells compared to stromal cells in most lesions. We also identified additional regions of cytokeratin-positive cells without a traditional luminal glandular structure. Often, these were in close proximity to other cytokeratin-positive glandular structures. It is possible they represent addition glandular structures not visible in the plane through which the sections were prepared or potential dysregulated epithelial growth. Their identification may assist in the detection of epithelial cells. Conversely, non-classical glandular-like structures may suggest a loss of structural organisation. Somatic mutations that confer a growth advantage have been observed in epithelial cells of endometriosis [11,12]. Further examination of these structures with specific markers may provide valuable prognostic information that could lead to the personalisation of treatments.

In this study, some lesions were removed from women that were receiving hormonal treatment. Hormonal treatments are believed to assist in the reduction of symptoms through a downregulation of systemic oestrogen concentrations. We have previously shown that hormones can influence stromal cell proliferation and inflammation in vitro [13,14,15] and there is the potential these treatments influence the lesions directly. We found no significant difference in the number of either epithelial or stromal cells in lesions from women who received either OCP or GnRHa. Although, interestingly, while the stromal cells from lesions excised from GnRHa-treated patients did not appear to be affected, it did appear epithelial cells were decreased, suggesting an epithelial-specific influence of this drug. Although the small GnRHa sample numbers limit the robustness of this interpretation, previous studies have also shown GnRHa can lead to apoptosis of endometrial epithelial cells [16].

The source of endometriosis pain remains an enigma. A focus on the microenvironment however, is making progress in elucidating the mechanisms involved. The proximity of peripheral nerve fibres at the lesion site has been related to the extent of pain experienced [17,18]. Inflammation stimulated by this ectopic tissue can provide a means of communication between the lesions and nerves [9]. Macrophages, at increased concentrations around endometriotic lesions [19], may promote innervation through oestradiol [20]. Macrophage-derived IGF-1 promotes sprouting neurogenesis and nerve sensitisation [21]. As the size of each endometriosis foci increases, the potential for them to come within sufficient proximity of a nerve fibre to mediate an influence also increases.

It has long been discussed whether the menstrual stage can Influence endometriotic lesion activity [22] through hormonal activation. A recent histological analysis indicated significant heterogeneity of SUP lesions that were not linked to the cycle stage [23] and a comprehensive review of the literature supported the view that endometriotic lesion characteristics do not reflect the menstrual stage [24] and that reduced responsiveness to hormones may be linked to the downregulation of progesterone receptor as lesions age [25]. Whether these influences will lead to changes in the cellular composition of individual lesions and whether they can be related to clinical outcomes has not yet been assessed, as it is limited by the ability of humans to identify, count, and quantitate every individual cell within a histological section. In this study, we were unable to assess the influence of the menstrual stage on the number of epithelial and stromal cells due to the limitation of numbers but believe that this could be greatly assisted by the use of machine learning approaches to automatically detect, quantify, and categorise different cell types identified in histological images, in addition to assisting identification between lesions and patient outcomes.

Finally, we compared the automated cell counts, as a proxy for lesion size and microenvironment, with patient symptoms. This analysis identified a positive correlation between the number of cells in each section and abdominal pain. This correlation was observed with both epithelial and stromal cells but was strongest when both were considered together. While this study is small, it indicates a potential for clinical symptoms to be linked to disease presentation through the computer-aided histological interpretation of surgically excised endometriotic lesions. Endometriosis shows a significant and diverse set of symptoms that can only be adequately addressed if treatment is personalised [26]. Even after surgical excision of the endometriotic lesion, patient management must continue, as up to 50% of patients will experience recurrence [27], with an increase in lesion severity [28]. Recent evidence suggests the recurrence can be limited with targeted treatment [29]. With the further application of machine learning and artificial intelligence to histological diagnosis, there is potential to improve the disease prognosis [30] and reduce the social and economic impact of endometriosis by tailoring treatment to individual outcomes. 

These results, of course, must be interpreted with caution, as it is yet to take into account the 3D dimensional scale of the lesions, the influence of different hormones on reported pain and lesion cells, or the timing of the disease. It is, however, a result of interest and in the context of the previous literature, we suggest that it is not simply lesion size but rather that more and larger endometriosis foci are more likely to be located within sufficient proximity to a sensory nerve [17], facilitating paracrine interaction through inflammation/hormonal mediators to both stimulate and sensitise the nerves.

In conclusion, this study, utilising computer software to quantitate cells, was able to agnostically and relatively easily quantitate endometriosis lesion size and cell content. It suggests that employing computer-aided histopathology may improve the potential lesion characteristics that can be related to diagnostic parameters and assist with prognostic information, direct patients to relevant adjuvant treatment, and facilitate a move towards personalised treatment.

## Figures and Tables

**Figure 1 jpm-12-01519-f001:**
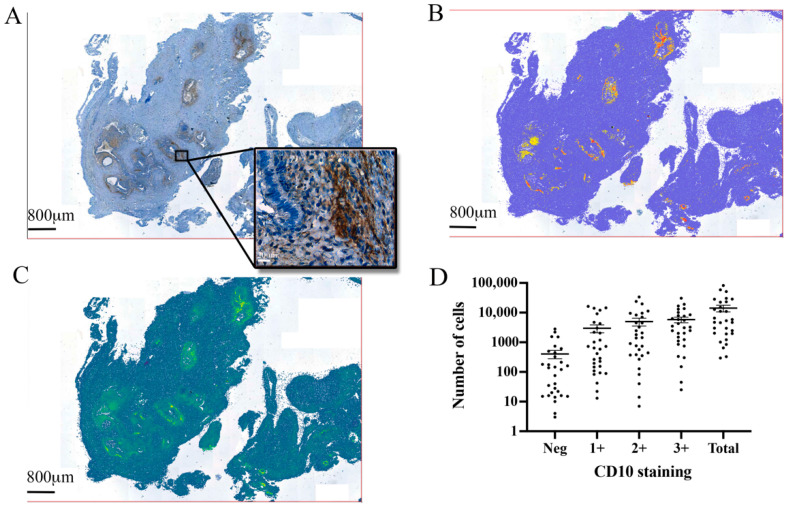
**CD10 and the automated identification of endometrial stromal cells in excised endometriotic tissue.** (**A**) Mouse anti-CD10 antibody-labelled endometrial stromal cells in the excised tissue, revealing multiple endometriosis foci. (**B**) Automated cell detection was performed with QuPath software over the entire excised lesion. (**C**) Pixel smoothing identified the size of each endometriosis foci, each individual cell within the lesion, and the relative staining intensity of each cell. (**D**) Total cell counts showed the number of stromal cells identified by automated software and the number of cells that were considered either negative for CD10 staining, lightly (1+), moderately (2+), or heavily (3+) immunoreactive for CD10.

**Figure 2 jpm-12-01519-f002:**
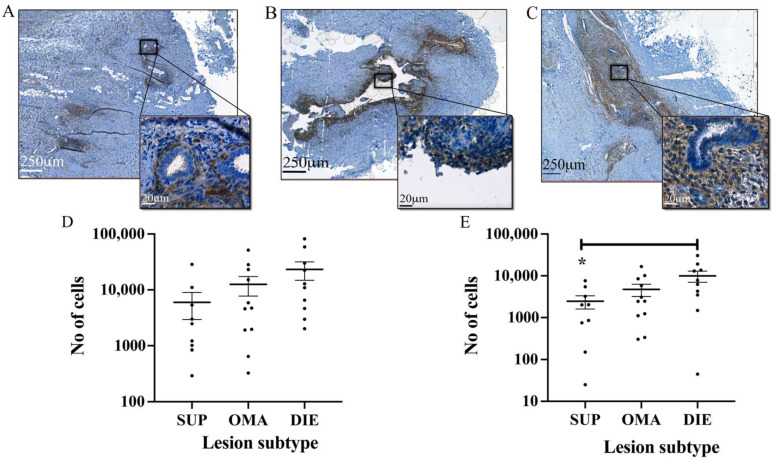
**CD10-positive stromal cells in endometriotic lesions of different subtypes.** The CD10 antibody and automated analysis of images using QuPath identified endometriotic stromal cells in lesions from (**A**) SUP, (**B**) OMA, and (**C**) DIE. (**D**) The total number of stromal cells was lowest in SUP lesions followed by OMA lesions. DIE lesions contained the largest number of stromal cells. (**E**) There were significantly more stromal cells with heavy CD10 staining intensity (3+) in DIE lesions compared to SUP lesions. * *p* < 0.05.

**Figure 3 jpm-12-01519-f003:**
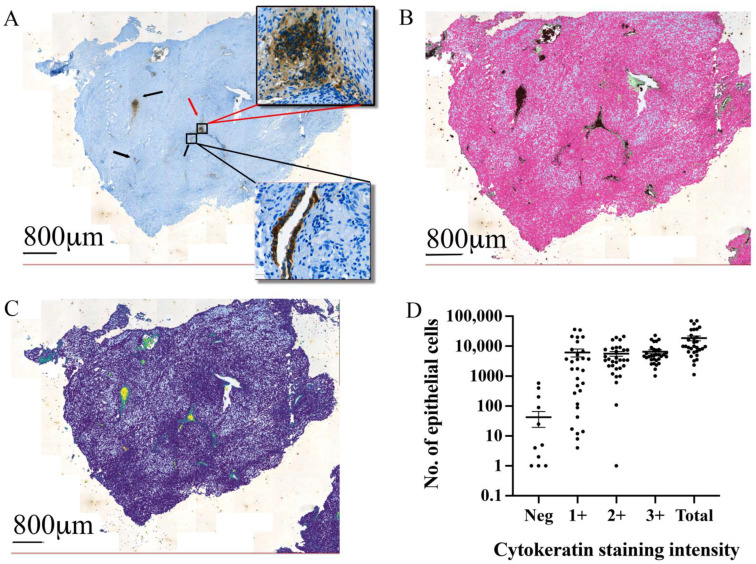
**Cytokeratin and automated identification of epithelial cells in endometriotic lesions.** (**A**) Mouse anti-cytokeratin antibody staining identified regions with characteristic glandular structures as epithelial cells (black arrows), and positive epithelial cells without the characteristic glandular structure (red arrows). (**B**) Automated cell detection using the object classifier quantified positive epithelial cells within the excised tissue. (**C**) Pixel smoothing provides the options to observe the extent of the epithelial compartment in the endometriotic lesions. (**D**) The automated counting of epithelial cells showed similar numbers of light (1+), moderate (2+), and heavily stained (3+) epithelial cells in all samples. Only a minimal number of automatically detected epithelial cells were negative for cytokeratin expression.

**Figure 4 jpm-12-01519-f004:**
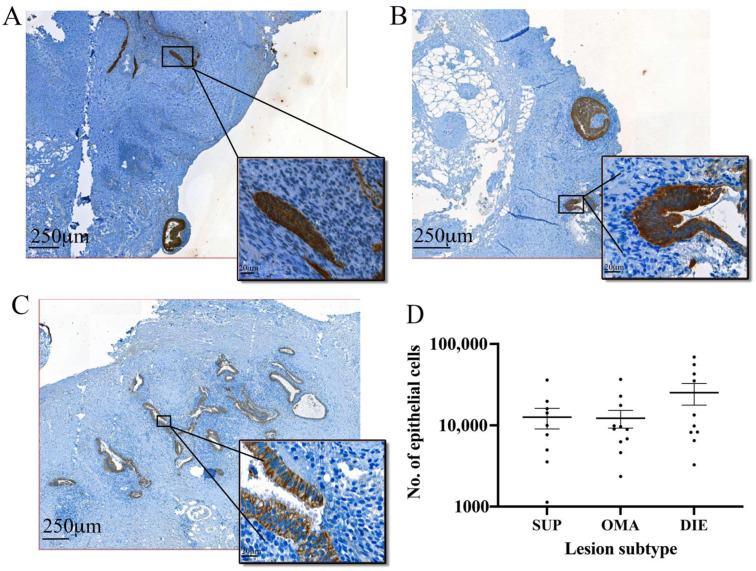
**Epithelial cell content in endometriotic lesions from different subtypes.** Using anti-cytokeratin antibody and automated machine learning to detect epithelial cells in (**A**) SUP, (**B**) OMA, and (**C**) DIE lesions, we found that (**D**) while there was a higher number of epithelial cells in DIE lesions, this difference was not significant.

**Figure 5 jpm-12-01519-f005:**
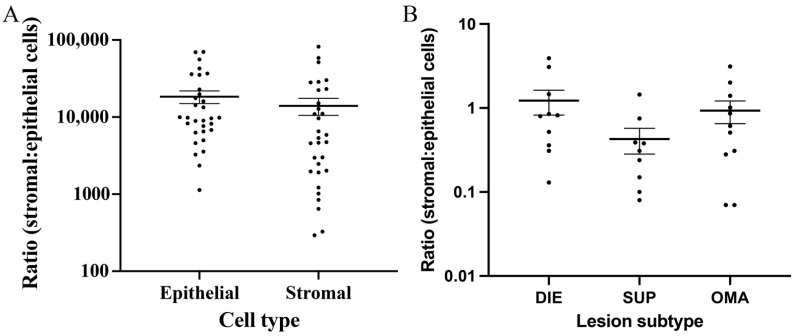
**Comparison between epithelial and stromal cell numbers in excised lesions.** (**A**) The number of epithelial and stromal cells in all endometriosis foci identified in the excised endometriosis tissue showed no significant difference, with approximately equal numbers of cells in both lesions. (**B**) There was no significant difference in the ratio of stromal:epithelial cells based on lesion subtypes. There was a significant correlation between the ratio of stromal to epithelial cells and dyspareunia.

**Figure 6 jpm-12-01519-f006:**
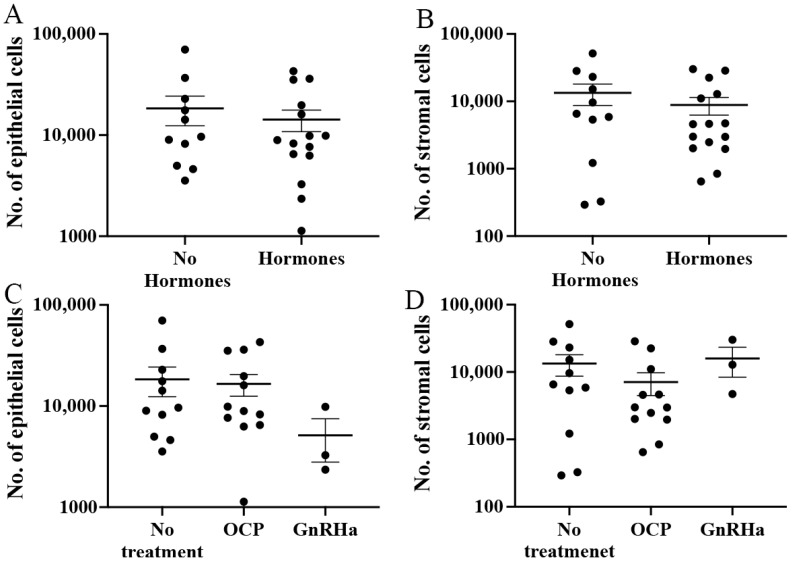
**Relationship between cell numbers and hormonal treatment.** A comparison between the number of (**A**) epithelial and (**B**) stromal cells found in samples derived from women who were not using hormonal treatments and women who were using hormonal treatments. Stratification of hormonal treatments via OCP and GnRHa did not show any significant difference in either (**C**) epithelial cells or (**D**) stromal cells.

**Table 1 jpm-12-01519-t001:** Clinical parameter of patients included in this study.

	SUP (Mean ± SEM)	OVA (Mean ± SEM)	DIE (Mean ± SEM)	* Total (Mean ± SEM)	*p*
*n*	**7**	**9**	**9**	**26**	
Age (years)	30.29 ± 1.46	33.04 ± 1.25	34.78 ± 2.06	32.68 ± 0.91	0.1523
BMI (kg/m^2^)	21.99 ± 1.40	21.53 ± 1.40	22.38 ± 1.09	21.93 ± 0.68	0.8902
Menstrual pain	4.60 ± 1.75	4.83 ± 1.66	8.00 ± 0.77	5.82 ± 0.85	0.2511
Abdominal pain	1.57 ± 0.84	3.57 ± 0.94	2.00 ± 0.82	2.50 ± 0.50	0.2576
Dsypareunia	0.86 ±0.70	3.71 ± 1.30	2.29 ± 0.99	2.23 ± 0.60	0.175

* Note: one additional lesion was from an unidentified region and therefore not included in the comparison between lesion types, SEM = standard error of the mean.

**Table 2 jpm-12-01519-t002:** Association between rAFS stage and cell numbers.

	rAFS Mild (I–II)		rAFS Severe (III–IV)		*p* Value
	Mean	SEM	Mean	SEM	
Epithelial cells	24,096	8195	16,454	3659	0.3368
Stromal cells	10,207	3702	15,336	4504	0.5267
Total cells	34,303	9483	31,791	7882	0.8642

SEM = standard error of the mean.

**Table 3 jpm-12-01519-t003:** Correlation between automated cell counts and pain symptoms.

	Menstrual Pain			Abdominal Pain			Dyspareunia		
	** *n* **	** *r* **	** *p* **	** *n* **	** *r* **	** *p* **	** *n* **	** *r* **	** *p* **
**Epithelial cells**	17	0.1521	0.5600	22	0.6066	*0.0028 ***	22	−0.1260	0.5763
**Stromal cells**	17	0.1216	0.6421	22	0.5117	*0.0149 **	22	0.2589	0.2447
**Total cells**	17	0.1683	0.5186	22	0.6782	*0.0005 ****	22	0.0573	0.7999

* *p* < 0.05, ** *p* < 0.01, *** *p* < 0.001.

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
