# Peer review of "Computer-Aided Histopathological Characterisation of Endometriosis Lesions"

_jpm, 2022, doi:10.3390/jpm12091519_

Round 1

Reviewer 1 Report

Thank you very much for the invitation to review of the manuscript. It a great pleasure for me.

The purpose of t McKinnon et al. was to check how using of histopathological software to identify and quantify the number of endometrial epithelial and stromal cells within excised endometriotic lesions can improve diagnostic of endometriosis and assess the relationship between the cell contents and lesion subtypes. That is very interesting paper, however I have only few questions:

1.      In the introduction, it is worth adding a clearly defined purpose of the work

2.      I am asking for better characteristics of the study group, in what phase of the menstrual cycle the laparoscopy was performed, were the patients treated pharmacologically before the procedure, did the patients have the stage of advanced disease assessed?

3.      What other symptoms were reported by patients?

4.      How many of them suffered from infertility and which from chronic pain?

5.      Was there a correlation between the severity of the disease? or oestradiol concentration?

Author Response

Reviewer 1:

  1. Thank you very much for the invitation to review of the manuscript. It a great pleasure for me.

Answer: NA

  1. The purpose of McKinnon et al. was to check how using of histopathological software to identify and quantify the number of endometrial epithelial and stromal cells within excised endometriotic lesions can improve diagnostic of endometriosis and assess the relationship between the cell contents and lesion subtypes. That is very interesting paper, however I have only few questions:

Answer: NA

  1. In the introduction, it is worth adding a clearly defined purpose of the work

Answer: We have added in a sentence at the end of the introduction section to clear define the purpose of our work “Using this technology we wished to determine whether automated histological software could provide additional data to aid the diagnosis and prognosis of endometriosis.

  1. I am asking for better characteristics of the study group, in what phase of the menstrual cycle the laparoscopy was performed, were the patients treated pharmacologically before the procedure, did the patients have the stage of advanced disease assessed?

Answer: Thank you for the comment we agree that this is important. In the patient data section in the results there is already information on the rAFS stage of disease and hormonal treatments taken by women.

Line 166, Stage of disease, “The revised American Fertility Score (rAFS) of the 26 women included 11 with stage IV, 10 with stage III, 3 with stage II and 2 women with stage

Line 167, Hormonal treatment prior to surgery “In this cohort 11 patients were receiving no hormonal treatments prior to surgery, 8 patients were receiving oral contraceptives, 3 patients reported GnRHa usage. In the remaining 4 patients we were unable to confirm their treatment history.”

We have also now included data to indicate the menstrual cycle stage at which surgery was performed in the results section, when available, and discussed the possible influence of the menstrual stage on endometriotic lesion characteristics in the discussion.

Results

Line 170” To determine the menstrual stage at which lesions were removed we relied on self-reported cycle day. Women taking hormonal treatments (n= 11) were considered amenorrheic, five women provided sufficient information to be considered post-luteal stage with no information available for the remaining 10 women.”

Discussion

Line 346” It has long been discussed whether the menstrual stage can influence endometriotic lesion activity (22)through hormonal activation. A recent histological analysis indicated significant heterogeneity of SUP lesions that were not linked to cycle stage (23) and a comprehensive review of the literature supported the view that endometriotic lesions characteristics do not reflect the menstrual stage (24) and that reduced responsiveness to hormones may be linked to the downregulation of progesterone receptor as lesions age (25). Whether these influences will lead to changes in the cellular composition of individual lesions and whether they can be related to clinical outcomes has not yet been assessed, as it is limited by the ability of humans to identify, count and quantitate every individual cell within a histological section. In this study, we were unable to assess the influence of the menstrual stage on the number of epithelial and stromal cells due to the limitation of numbers, but believe that this could be greatly assisted by using machine learning approaches to automatically detect, quantify and categorise different cell types identified in histological images, as well as assisting the identification between lesions and patient outcomes.

  1. What other symptoms were reported by patients?

Answer: We thank the reviewer for the interesting question. In our presurgical questions, we did not directly ask about any additional symptoms from these patients. The focus of this study was to generate a method to automate the detection of endometriotic cells in endometriosis lesions. Now that we have built a method to determine the size of the lesions it would be very useful in the future to relate these lesion characteristics to additional clinical symptoms and we hope to do this soon.

  1. How many of them suffered from infertility and which from chronic pain?

Answer: We thank the reviewer for an interesting question. As mentioned above we think this will be a good potential to relate lesion characteristics to symptoms unfortunately in our research database we do not have information about the fertility status of these patients.

  1. Was there a correlation between the severity of the disease? or oestradiol concentration?

Answer: This is an interesting question. We have additionally performed an analysis with lesion severity. This analysis showed no correlation with disease severity as assessed via the revised American fertility society (rAFS) score. This has been included in the text as Table 2.

Unfortunately, we did not have oestradiol measurements for these patients and are unable to do this additional work. We do believe that this is a good idea and would like to introduce this measurement into future studies.

Reviewer 2 Report

1. The study group includes only 26 patients. Taking into account the heterogeneity of endometriosis it is too few cases to draw adequate conclusions, especially when the Authors consider only three cases (3 patients reported GnRHa usage).
As the Authors, themselves emphasize endometriotic lesions are extremely heterogenic, therefore in my opinion the study group should include at least 60 cases.
2.
 There is no control group in the study?
3.
Please explain the abbreviations when you use them for the first time, for example in line 43 of the Abstract ( DIE lesions compared to SUP and OMA lesions).
4.
  I strongly recommend the Authors check the relationship between total endometrial cells and the clinical stages of endometriosis according to American Fertility Score.
5. No statistical significance should be noted when stating that the p-value was above 0.05.

6. Please change the way of recording statistical significance in the manuscript as follows: p < 0.05, p <0.01, p <0.001, or p <0.0001 respectively.

Author Response

Reviewer 2:

  1. The study group includes only 26 patients. Taking into account the heterogeneity of endometriosis it is too few cases to draw adequate conclusions, especially when the Authors consider only three cases (3 patients reported GnRHa usage).

Answer: We agree with the reviewer that endometriosis is extremely heterogeneous and researchers must be careful to not over-interpret the results. We believe the power of this manuscript is the description of the new technology and how its application could be applied to endometriosis diagnosis and prognosis, rather than identifying specific differences in clinical parameters. We believe it is important to show that this technology can be used effectively before applying it to a cohort of sufficient size that can match lesion characteristics to clinic parameters.

This study highlighted the heterogeneity of endometriosis by comparing subtypes and showing that the cellular content in fact also varies. We have however been very careful not to over-interpret the results. While we agree sample size for the comparison of GnRHa treatment is small, and we report no significant difference between GnRHa usage and no treatment we believe it is important to report such results and hope that they can be followed up in a larger study targeted specifically at assessing the influence on hormones on endometriosis lesions.

  1. As the Authors, themselves emphasizeendometriotic lesions are extremely heterogenic, therefore in my opinion the study group should include at least 60 cases. 

Answer: As we have indicated in the above response the main focus of this study was to assess the potential to apply computer-aided cellular identification in endometriosis diagnosis. We believe we have been able to show the potential of this technique and hope that it will prove useful in identifying significant variations in endometriotic lesions that may be related to clinical symptoms. In such cases the heterogeneity of endometriosis, and variation in treatment options these groups should be carefully selected, requiring significant time to collect samples and perform the relevant analysis. We believe these should be done in the future and this manuscript is vital in setting the template at which this can be applied.

  1. There is no control group in the study?

Answer: We have not included a control group without endometriosis as in such a case no lesion will be excised and it will not possible to identify endometriotic lesions, nor for the software to detect endometriotic cells.

  1. Please explain the abbreviations when you use them for the first time, for examplein line 43 of the Abstract ( DIE lesions compared to SUP and OMA lesions). 

Answer: We thank the reviewer for noticing this mistake and have used the full name in the abstract and explained the abbreviation in their first use in the introduction (2nd paragraph)

  1. I strongly recommend the Authors checkthe relationship between total endometrial cells and the clinical stages of endometriosis according to American Fertility Score. 

Answer: As requested we have compared the relationship between rAFS staging split between mild (I-II) and severe (III-IV) disease and the number of cells in endometriotic lesions. The results of this analysis have now been included in the manuscript in Table 3 and in the results section. We found no significant association.

Line 266 “Finally, we compared automated cell quantification values to patient symptoms recorded prior to surgery and endometriosis staging of patients recorded during surgery. We found no association between stage of disease, separated into either mild( rAFS stage I-II) or (severe III-IV) for either epithelial,  stromal cells or cells combined (Table 2).”

  1. No statistical significance should be noted when stating that thep-value was above 0.05.

Answer: As requested and in line with the request below we have removed any p values that were not significant

  1. Please change the way of recording statistical significance in the manuscript as follows:p < 0.05, p <0.01, p <0.001, or p <0.0001 respectively.

Answer: We have recorded the statistical significance as requested and replaced the actual p-value with the categories of significance that have been requested.

Reviewer 3 Report

The authors have described a novel technique of computer-aided histopathological characterization in endometriosis lesions. The study has been well conducted and the manuscript is well written. However, several corrections should be made to achieve better comprehension. First of all, the title of the manuscript should be simplified as "Computer-aided characterization of endometriosis lesions". Second, the abbreviations should be initially written in open form, then followed by abbreviations in parentheses. Third, the discussion subheading should be added appropriately to separate results from the conclusion part. Lastly, the references that were published before 2007 should be replaced with newer and more up-to-date ones if possible. 

Author Response

Reviewer 3:

  1. The authors have described a novel technique of computer-aided histopathological characterization in endometriosis lesions. The study has been well conducted and the manuscript is well written. However, several corrections should be made to achieve better comprehension. First of all, the title of the manuscript should be simplified as "Computer-aided characterization of endometriosis lesions".

Answer: Thank you for the suggestion. We have amended the title as requested.

  1. Second, the abbreviations should be initially written in open form, then followed by abbreviations in parentheses.

Answer: Thank you for the comment we have adjusted any abbreviation as requested.

  1. Third, the discussion subheading should be added appropriately to separate results from the conclusion part.

Answer: A discussion subheading has been included on Line 287.

  1. Lastly, the references that were published before 2007 should be replaced with newer and more up-to-date ones if possible. 

Answer: Where appropriate new reference have been added.

Reviewer 4 Report

Dear Editor, thank you for the opportunity to review the manuscript entitled: ''Computer-aided histopathological characterisation of endometriosis lesions; potential to enhance prognostic value''. 

Firstly, this title of manuscript has to be changed in order to improve scientific soundness, current title is not attractive enough. 

Introduction has to be more updated regarding references so I suggest use recently published articles such as (PMID: 34718292). 

Methodology - I am only suggesting further clarification of inclusion/exclusion criteria, are patients consecutively sampled or are there any other criteria? 

In Discussion section it is important to address strengths and limitations of this study and further perspectives, including potential clinical benefits of this approach... please estimate cost-benefit ratio in everyday clinical work.

Author Response

Reviewer 4:

  1. Dear Editor, thank you for the opportunity to review the manuscript entitled: ''Computer-aided histopathological characterisation of endometriosis lesions; potential to enhance prognostic value''. 
  2. Firstly, this title of manuscript has to be changed in order to improve scientific soundness, current title is not attractive enough. 

Answer: The title has been changed in line with the request of Reviewer 3.

  1. Introduction has to be more updated regarding references so I suggest use recently published articles such as (PMID: 34718292). 

Answer: The reference has been added in an additional section in the discussion that addressed the cost benefits that could be achieved by automating diagnosis. Line 368.

  1. Methodology - I am only suggesting further clarification of inclusion/exclusion criteria, are patients consecutively sampled or are there any other criteria?

Answer: Patients were not consecutively sampled. These were collected as part of a larger study that was mainly focussed on peritoneal sample and lesions were only collected in some cases where the surgeon was willing and capable.

  1. In Discussion section it is important to address strengths and limitations of this study and further perspectives, including potential clinical benefits of this approach... please estimate cost-benefit ratio in everyday clinical work.

Answer: We agree with the reviewer that strengths and limitations should be addressed, this has been done in the discussion section. We have additionally added a qualitative discussion on the potential cost-benefit analysis. As our team are not qualified health economist we have not conducted a formal quantification of cost-benefit analysis.

Line 365 “While this study is small it indicates a potential for clinical symptoms to be linked to disease presentation through the computer-aided histological interpretation of surgically excised endometriotic lesions. Endometriosis shows a significant and diverse set of symptoms that can only be adequately addressed if treatment is personalised (26). Even after surgical excision of the endometriotic lesion patient management must continue, as up to 50% of patients will experience recurrence (27) with an increase in lesions severity (28). Recent evidence suggests the recurrence can be limited with targeted treatment (29). With the further application of machine learning and artificial intelligence to histological diagnosis, there is potential to improve the disease prognosis (30) and reduce the social and economic impact of endometriosis by tailoring treatment to individual outcomes.”

Reviewer 5:

  1. Find your research very valuable and well-written. I have several minor comments to make:

Answer: Thank you for the positive comments.

  1. Please complete Table 1 by adding the units of measure (such as years, kg/m2, etc) in the first column, and by stating mean and standard deviation in the heading of following columns.

Answer: We have completed the requested changes.

  1. I would rename the Results chapter as Results and Discussions.

Answer: To remain consistent with the request from Reviewer 3 we would suggest keeping the results and discussion sections separated.

  1. Perhaps it would look better if the Conclusions were an independent chapter, rather than the last paragraph of the results.

Answer: The conclusion paragraph in the last section is part of the discussion and not the results.

  1. The number of references is quite reduced and the most recent reference dates 2019, while the others are older than 2016. I suggest an update with newer studies.

Answer: As part of additional discussion we have included additional references and where appropriate we have updated the reference list. In some cases we have tried to identify the original research papers and cite these rather than relying on more recent reviews or results.

Reviewer 5 Report

Esteemed Authors,

I find your research very valuable and well-written. I have several minor comments to make:

- please complete Table 1 by adding the units of measure (such as years, kg/m2, etc) in the first colum, and by stating mean and standard deviation in the heading of following columns.

- I would rename the Results chapter as Results and Discussions.

- Perhaps it would look better if the Conclusions were an independent chapter, rather than the last paragraph of the results.

- The number of references is quite reduced and the most recent reference dates 2019, while the others are older than 2016. I suggest an update with newer studies.

Author Response

Reviewer 5:

  1. Find your research very valuable and well-written. I have several minor comments to make:

Answer: Thank you for the positive comments.

  1. Please complete Table 1 by adding the units of measure (such as years, kg/m2, etc) in the first column, and by stating mean and standard deviation in the heading of following columns.

Answer: We have completed the requested changes.

  1. I would rename the Results chapter as Results and Discussions.

Answer: To remain consistent with the request from Reviewer 3 we would suggest keeping the results and discussion sections separated.

  1. Perhaps it would look better if the Conclusions were an independent chapter, rather than the last paragraph of the results.

Answer: The conclusion paragraph in the last section is part of the discussion and not the results.

  1. The number of references is quite reduced and the most recent reference dates 2019, while the others are older than 2016. I suggest an update with newer studies.

Answer: As part of additional discussion we have included additional references and where appropriate we have updated the reference list. In some cases we have tried to identify the original research papers and cite these rather than relying on more recent reviews or results.

Round 2

Reviewer 4 Report

N/A